# LATENT LIGHT SOURCE MODELING FOR SCENE RECONSTRUCTION UNDER DYNAMIC ILLUMINATION

## ABSTRACT

Modeling scenes under unknown, varying single-point illumination is crucial for applications such as interactive relighting, augmented reality, and robotics. However, existing dynamic novel-view synthesis methods struggle with changing lighting, often misinterpreting illumination-induced appearance variations (e.g., cast/self-shadows), which complicates optimization and degrades reconstruction quality. In this paper, we present a point-based framework that explicitly models dynamic scenes under rapidly changing single-point light sources, enabling accurate reconstruction of illumination effects and realistic novel-view rendering under arbitrary new lighting conditions. Our approach builds on 2D Gaussian splatting, augmenting each Gaussian splat with learned BRDF parameters and leveraging physics-based differentiable rendering to decouple reflectance and illumination for each point. Experiments on dynamic scenes demonstrate that our method faithfully reproduces complex lighting and shadow variations in novel views.

## 1 INTRODUCTION

Novel-view synthesis (NVS) has emerged as a central challenge in 3D vision, underpinning applications in virtual reality, augmented reality, robotics, and cinematic content creation. Recently, a series of progresses have been made in this field, such as NeRF (Mildenhall et al., 2021), 3DGS (Kerbl et al., 2023) and 2DGS (Huang et al., 2024). However, most of these works assumes the light source is fixed and ignore the influence caused by illumination variation such as the location change of the light source, which is ubiquitous in real world environments.

To address the reconstruction of dynamic light-source scenes, as shown in Figure 1(a), a possible solution is to adopt a relightable scene reconstruction framework trained on OLAT dataset Debevec et al. (2000). However, these methods Kuang et al. (2024); Liu et al. (2024b) require the prior knowledge (such as locations and intensity) of the light, as shown in Figure 1(b). It is costly even unavailable to record the light source in realistic applications. A related direction is neural inverse rendering under fixed or unknown but *time-invariant* illumination, which jointly estimates geometry and BRDF from multi-view images while recovering a single environment map (e.g., NeRFactor; TensoSDF) (Zhang et al., 2021; Li et al., 2024a). These approaches rely on photometric consistency under fixed lighting and typically assume far-field, low-frequency illumination. In our setting, a moving near-field point light changes the illumination at each timestamp, producing pronounced color and shadow variations that violate these assumptions. Therefore, neither OLAT-based relighting nor inverse rendering under fixed illumination can be directly applied to our scenario.

Another possible way is to model the task as the dynamical scene reconstruction Pumarola et al. (2021), and solves it by existing methods such as 4DGS Wu et al. (2024) or ED3DGS Katsumata et al. (2024a). However, these approaches primarily focus on modeling moving objects rather than time-varying lighting. As shown in Fig. 1(c), these methods may fail when lighting-driven color changes are mistaken for moving objects. One simple example is that existing dynamic reconstruction methods often treat shadows as moving objects, spawning temporally shifting "shadow Gaussians" around the scene. These shadow Gaussians, together with the normal Gaussians, participate in training and render visually plausible images: At a same timestamp, a shadow Gaussian produces a dark cast region, and it will generate the correct shadow with the normal Gaussians under one viewpoint. However, under another viewpoint, the same shadow Gaussian adopts object or back-

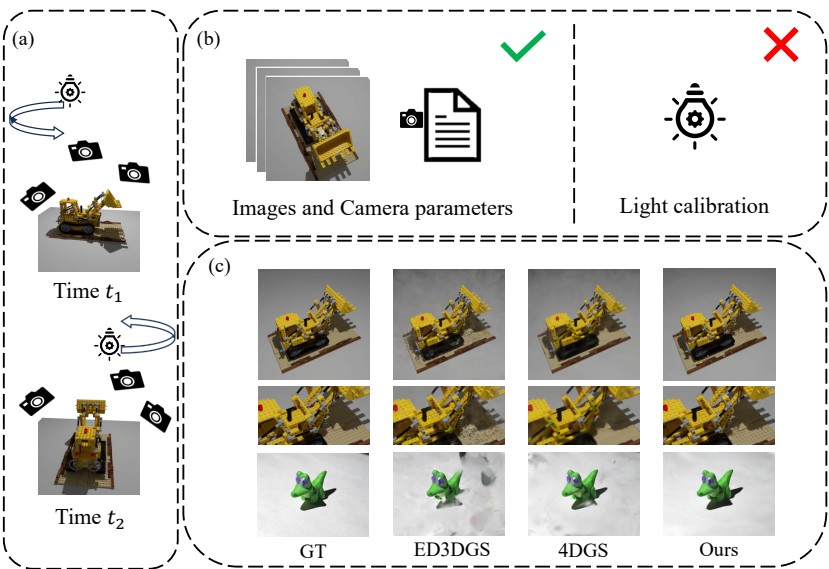

Figure 1: (a) Dynamic illumination scene: a single point light source continuously orbits the object, producing ever-changing lighting and shadow patterns. (b) Same as standard dynamic-scene reconstruction: only images with the corresponding camera intrinsics and extrinsics; no light-source information or calibration is required. In contrast to OLAT light-stage setups, no per-light direction, intensity, timing, or any other light-source calibration/metadata is provided or required. (c) On both the synthetic Lego and real datasets, 4DGS and ED3DGS blur shadow boundaries or miscolor unstable Gaussians, whereas our method delivers sharp shadows, accurate colors, and faithful reconstructions.

ground colors yet again yields a plausible composite. Consequently, the shadow Gaussians interfere with the normal Gaussians' ability to learn accurate geometry and color in shadowed areas, leading to blurred or incorrectly colored reconstructions (e.g., the distorted Lego result in Fig. 1(c)) or to erroneous point clouds showing background or object colors in real world scenes (also in Fig. 1(c)).

To address these challenges, we argue that it should model the light source directly to calculate the lighting-driven color changes rather than model shadows by Gaussians, where the latter causes additional learning difficulty. Motivated by Relightable 3D Gaussians (Gao et al., 2024), the bidirectional reflectance distribution function (BRDF) is used to render the images. Considering the case that the light source is unknown, a novel method is proposed based on latent light source modeling. Specifically, our method is designed based on 2DGS Huang et al. (2024), and introduces a two-branch neural network to model the latent light source: (1) one branch takes the timestamp as input to predict the location and intensity of the latent light source, and then these predictions are used, together with each Gaussian position, to calculate the light intensity reaching each Gaussian without occlusion. (2) The other branch takes the timestamp and the Gaussian positions as input to predict, for each Gaussian at that timestamp, its visibility and the environmental illumination it receives. Finally, we jointly calculate the actual incident light intensity for each Gaussian by combining the light intensity, visibility, and environmental illumination. Using each Gaussian's BRDF parameters, we then apply the rendering equation (Kajiya, 1986) to determine its color. In summary, our main contributions are as follows:

- A novel latent light-source modeling is introduced to tackle dynamic single-source scene reconstruction.

- We propose an illumination dual-branch MLP that jointly learns the latent light source and the Gaussians' BRDF parameters, enabling accurate color rendering under dynamic illumination.

- Extensive experiments on both synthetic and real world datasets indicate that our method significantly outperforms the baselines such as 4DGS and 2DGS.

## 2 RELATED WORKS

### 2.1 SCENE RECONSTRUCTION

Neural Radiance Fields (NeRF) (Mildenhall et al., 2021) represent scenes as continuous implicit functions optimized for static novel-view synthesis. Pioneering works (Park et al., 2021a;b; Pumarola et al., 2021) have broken this static limitation by extending NeRF to dynamic scenarios—introducing learnable deformation fields to model time-varying geometry. Subsequent researches (Fang et al., 2022; Cao & Johnson, 2023; Fridovich-Keil et al., 2023) have focused on accelerating dynamic NeRF training through explicit scene representations and more efficient optimization strategies to better capture temporal changes.

3D Gaussian Splatting (3DGS) Kerbl et al. (2023) has emerged as a powerful point-based rendering paradigm, offering real-time performance, fast convergence, and high-fidelity image quality. To extend its advantages to dynamic scenes, 4DGS (Wu et al., 2024), D3DGS Yang et al. (2024), ADC-GS Huang et al. (2025) and DG-mesh Liu et al. (2024a) introduces a learnable deformation field into the 3DGS pipeline to account for temporal variations in object geometry. In a related direction, several apporaches (Duan et al., 2024; Yang et al., 2023; Katsumata et al., 2024b) generalize the 3D Gaussian formulation to a time-aware 4D representation, explicitly modeling geometry changes across both space and time. These approaches have demonstrated strong performance in capturing complex motion patterns in dynamic environments. Building on the precise surface reconstruction of 2D Gaussian splatting (2DGS), recent works analogously incorporate temporal deformation. The method (Wang et al., 2024) embeds a learnable motion field into the 2DGS pipeline, enabling per-frame surface modeling with sub-millimeter accuracy. Likewise, the method (Zhang et al., 2024a) learns keypoint deformations on top of the 2DGS framework to reconstruct animated scenes. These approaches focus on reconstructing scenes with dynamic objects. However, scenes with moving objects are not the same as scenes with dynamic illumination. Modeling the shadow as Gaussians may cause additional learning difficulty, and these methods may fail when lighting-driven color changes are mistaken for moving objects, as shown in Fig. 1.

### 2.2 ILLUMINATION IN NEURAL SCENE REPRESENTATIONS

Early NeRF-based relighting and reflectance decomposition methods demonstrate powerful disentanglement of geometry, material properties, and lighting—for example, NeRD (Boss et al., 2021a) recovers per-point BRDFs and visibility fields, NRTF (Lyu et al., 2022) models global illumination via learned transfer operators, NeRFactor (Zhang et al., 2021) jointly factorizes illumination, shape, and reflectance without calibrated probes, and Tensoir's tensor decomposition captures complex light–material interactions (Jin et al., 2023). Parallel Gaussian-splatting frameworks recover spatially varying reflectance under fixed lighting: SVG-IR (Sun et al., 2025) uses spatially-varying Gaussians, GI-GS (Chen et al., 2024) and Relightable 3D Gaussians (Gao et al., 2024) decomposes global illumination across Gaussian primitives, IRGS traces 2D Gaussians under assumed light geometry (Gu et al., 2025), PRTGS precomputes radiance transfer for real-time relighting (Guo et al., 2024), GaussianShader integrates learned shading into Gaussians (Jiang et al., 2024), and static-scene global-illumination approximations (Wu et al., 2025). These methods all tackle the problem of disentangling illumination from geometry to perform relighting. However, they operate under the assumption of static illumination, and thus their approaches cannot be directly applied to our scenario.

For dynamic single-point lighting, several Gaussian-splatting relighting methods support varying illumination but still require explicit light parameters or calibrated cues: OLAT Gaussians embed per-point BRDFs yet depend on known light positions and intensities (Kuang et al., 2024); BIGS introduces bidirectional Gaussian primitives for view- and light-dependent effects (Liu et al., 2024b); Dihlmann et al. incorporate measured subsurface scattering into splatting (Dihlmann et al., 2024); DarkGS learns neural illumination for low-light exploration under assumed geometry (Zhang et al., 2024b); GS3 achieves efficient relighting via triple Gaussian splatting with light-probe data (Bi et al., 2024); and Neural-PIL pre-integrates incident illumination into analytic bases, necessitating extra lighting cues (Boss et al., 2021b). These methods excel in static or controlled settings but cannot generalize to truly time-varying lighting without prior knowledge. In contrast, **our framework requires no prior knowledge of light source's location and intensity.** We infer single point light parameters automatically, and augment each Gaussian with learned BRDF parameters.

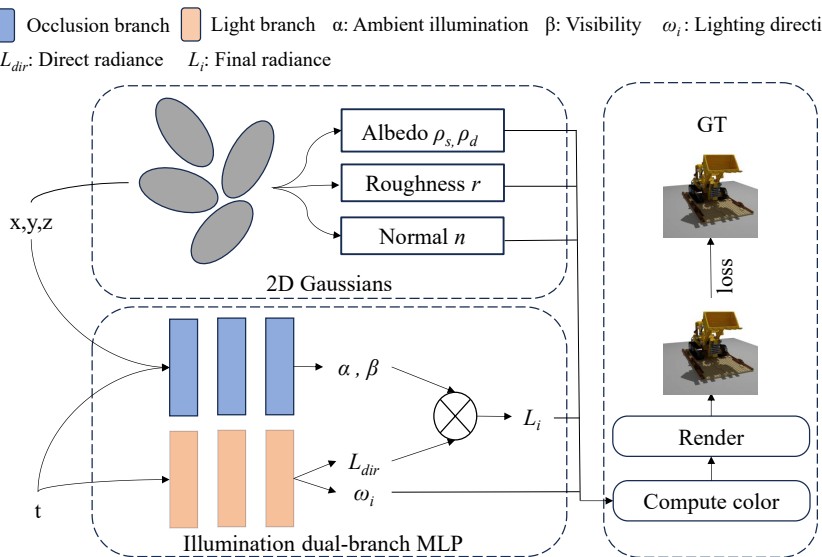

Figure 2: Overview of our framework. We augment each 2D Gaussian primitive with 3 learnable Disney BRDF parameters (specular albedo $\rho_s$, diffuse albedo $\rho_d$, and roughness $r$) and integrate a dual-branch neural network into the standard 2DGS pipeline to predict the lighting direction and intensity for each Gaussian. Whereas the original 2DGS uses spherical harmonics coefficients to compute color, we directly compute reflected radiance from the predicted parameters using rendering Equations (4), (5), and (6).

## 3 PRELIMINARY

### 3.1 2D GAUSSIAN SPLATTING

2DGS (Huang et al., 2024) is an explicit 3D scene representation that project each Gaussian point from three dimensions down to two, using an elliptic disk rather than an ellipsoid as the primitive. Each 2D Gaussian is defined by the parameters $G = \{\mathbf{p}, \mathbf{t}_u, \mathbf{t}_v, s_u, s_v, o, \mathbf{c}\}$, where $\mathbf{p}$ is the center of the ellipse, $\mathbf{t}_u$ and $\mathbf{t}_v$ are the two principal tangential vectors with the normal defined as $\mathbf{t}_w = \mathbf{t}_u \times \mathbf{t}_v$, giving the rotation matrix $\mathbf{R} = [\mathbf{t}_u, \mathbf{t}_v, \mathbf{t}_w]$, $s_u$ and $s_v$ are the scale factors packed into the scale matrix $\mathbf{S} = \mathrm{diag}(s_u, s_v, 0)$, $o$ is the opacity, and $\mathbf{c}$ is the RGB color. Accordingly, we also write $G = \{\mathbf{p}, \mathbf{R}, \mathbf{S}, o, \mathbf{c}\}$. When rendering, the color $C$ of pixel is computed by:

$$C = \sum_{i \in \mathcal{N}} \mathbf{c}_i \, \alpha_i \prod_{j=1}^{i-1}(1 - \alpha_j), \tag{1}$$

where $\mathcal{N}$ is the ray's sorted list of intersected Gaussians, $\mathbf{c}_i$ is the $i$-th Gaussian's color, and $\alpha_i$ is its opacity.

The local Gaussian coordinates of the intersection point $\mathbf{x}$ are given by:

$$u(\mathbf{x}) = \frac{(\mathbf{x} - \mathbf{p})^\top \mathbf{t}_u}{s_u}, \quad v(\mathbf{x}) = \frac{(\mathbf{x} - \mathbf{p})^\top \mathbf{t}_v}{s_v}, \tag{2}$$

and then the opacity at $\mathbf{x}$ is:

$$\alpha(\mathbf{x}) = o \exp\left(-\left[u(\mathbf{x})^2 + v(\mathbf{x})^2\right]\right). \tag{3}$$

### 3.2 BRDF

The bidirectional reflectance distribution function (BRDF) $f_r(\mathbf{x}, \omega_i, \omega_o)$ characterizes how light arriving at point $\mathbf{x}$ from direction $\omega_i$ is scattered into direction $\omega_o$, encapsulating a material's reflectance properties. It serves as the core kernel of the rendering equation, which integrates the

BRDF over the hemisphere of incoming directions to compute the total outgoing radiance. The rendering equation (Kajiya, 1986) is given by:

$$L_o(\mathbf{x}, \omega_o) \;=\; \int_\Omega f_r(\mathbf{x}, \omega_i, \omega_o)\, L_i(\mathbf{x}, \omega_i)\, (\mathbf{n}\cdot\omega_i)\, \mathrm{d}\omega_i \,, \tag{4}$$

where $L_o(\mathbf{x}, \omega_o)$ is the radiance leaving point $\mathbf{x}$ toward direction $\omega_o$, $L_i(\mathbf{x}, \omega_i)$ is the incoming radiance from direction $\omega_i$, $\mathbf{n}$ is the unit surface normal, and $\Omega$ denotes the hemisphere above $\mathbf{x}$. The dot product $(\mathbf{n}\cdot\omega_i)$ accounts for foreshortening, and $\mathrm{d}\omega_i$ is the differential solid angle. In our work, we use a simplified Disney BRDF (Burley & Studios, 2012):

$$f_r(\mathbf{x}, \omega_i, \omega_o) \;=\; f_d + f_s(\omega_i, \omega_o). \tag{5}$$

The diffuse term $f_d = \rho_d/\pi$ and the specular term is:

$$f_s(\omega_i, \omega_o) \;=\; \frac{\rho_s\, D\big(h;\, r\big)\, F(\omega_i,\, h)\, G(\omega_i, \omega_o;\, r)}{4\, (n\cdot\omega_i)\, (n\cdot\omega_o)} \,, \tag{6}$$

where $\rho_d \in [0,1]^3$ is the diffuse albedo, $\rho_s \in [0,1]^3$ is the specular albedo at normal incidence, $r \in [0,1]$ is the surface roughness, and $h = (\omega_i+\omega_o)/\|\omega_i+\omega_o\|$ is the half-vector (Cook & Torrance, 1982). Here, $D(h; r)$ denotes the GGX (Walter et al., 2007) normal distribution function, $F(\omega_i, h)$ is Schlick's Fresnel approximation (Schlick, 1994), and $G(\omega_i, \omega_o; r)$ is the Smith masking–shadowing geometry term (Smith, 1967).

## 4 METHOD

For computational simplicity, let $\omega_i$ denote the direction from the Gaussian center $\mathbf{p}$ to the light source position, and $\omega_o$ denote the direction from $\mathbf{p}$ to the camera position. Our overall architecture is depicted in Figure 2.

### 4.1 BRDF-PARAMETERIZED 2DGS UNDER DYNAMIC ILLUMINATION

Our goal is to reconstruct scenes under dynamic single source illumination without the prior knowledge of light source. A naive alternative is to fit a time-conditioned spherical-harmonics (SH) color field to explain per-timestamp appearance changes. However, unlike static captures with hundreds of views, dynamic captures provide only a handful of views per timestamp, leaving the per-timestamp SH coefficients underconstrained and harming generalization to novel viewpoints (see Appendix A for a simple test). Therefore, to model color variations while preserving cross-view consistency, inspired by Gao et al. (2024), we assign each 2D Gaussian a compact, physically based material model with three learnable Disney BRDF parameters—roughness $r$, specular albedo $\rho_s$, and diffuse albedo $\rho_d$. Once the Gaussians have learned the correct BRDF attributes, and given the per-timestamp illumination predicted by our illumination dual-branch mlp, per-view colors are computed according to Equation (4).

### 4.2 ILLUMINATION DUAL-BRANCH MLP

A straightforward alternative is to train a single MLP that, at each timestamp $t$, regresses for every 2D Gaussian its incident direction and intensity. However, because appearance depends jointly on direction and intensity through the BRDF, per-Gaussian direction regression tends to absorb variations that should be explained by material, biasing BRDF learning and degrading generalization to novel views. Moreover, using a single "intensity" channel to encode both ambient illumination and visibility, increasing ambiguity and reducing prediction quality. Therefore, built on the standard 2DGS rasterization pipeline, we introduce a dual-branch illumination network that (a) infers a global latent light trajectory and emission and (b) predicts per-point visibility and an ambient residual that captures soft indirect effects.

**The light branch.** Denoted by $f_\theta$, the light branch takes only the current timestamp $t$ as input and regresses the 3D coordinates of the point-light source $\mathbf{p}_l(t) \in \mathrm{R}^3$ together with its emission intensity $E_l(t) \in \mathrm{R}^+$:

$$f_\theta(t) = \big(\mathbf{p}_l(t),\, E_l(t)\big). \tag{7}$$

Once we have estimated the position of the light source $\mathbf{p}_l(t)$ and the intensity of emission $E_l(t)$ at time $t$, we can compute, for each 2D Gaussian, the incident lighting direction and the corresponding direct illumination intensity. Under the ideal point-source assumption (neglecting occlusion), the direct radiance $L_{\text{dir}}$ reaching a Gaussian at position $\mathbf{p}$ decays according to the inverse-square law:

$$L_{\text{dir}}(\mathbf{p}, t) = \frac{E_l(t)}{\|\mathbf{p}_l(t) - \mathbf{p}\|^2} \, . \tag{8}$$

This expression captures the physical principle that intensity decays proportionally to $1/r^2$, where $r = \|\mathbf{p}_l(t) - \mathbf{p}\|$ is the distance to the light source. That is, we could predict the lighting direction $\omega_i$ and direct illumination intensity for each 2D Gaussian by the light branch.

**The occlusion branch.** The occlusion branch $g_\phi$ receives the Gaussian position $\mathbf{p}$ and timestamp $t$ as input. To capture high-frequency variations, inspired by NeRF (Mildenhall et al., 2021), we apply positional encoding $\gamma(\mathbf{p})$ and $\gamma(t)$. The concatenated encodings are passed through $g_\phi$, which produces a feature embedding that is decoded by two tiny decoders into the visibility $\beta(\mathbf{p}, t) \in [0, 1]$ and ambient illumination $\alpha(\mathbf{p}, t) > 0$. $\beta(\mathbf{p}, t) = 1$ indicates full visibility, while $\beta(\mathbf{p}, t) = 0$ denotes complete occlusion

**Illumination assembly.** Having obtained from the light branch the incident lighting direction and direct illumination intensity for each 2D Gaussian, and from the occlusion branch the visibility and ambient illumination, we determine the final radiance for each Gaussian. Considering that ambient illumination compensates for occluded regions and captures indirect lighting effects to ensure smooth and physically plausible shading, we define the final illumination as:

$$L_i(\omega_i, \mathbf{p}, t) = L_{\text{dir}}(\mathbf{p}, t)\,\beta(\mathbf{p}, t) + \alpha(\mathbf{p}, t). \tag{9}$$

The direct lighting term $L_{\text{dir}}(\mathbf{p}, t)$ is attenuated by the learned visibility $\beta(\mathbf{p}, t)$ and supplemented by the ambient component $\alpha(\mathbf{p}, t)$ to model any residual illumination.

### 4.3 Color Computation and Rasterization

In our pipeline, for each 2D Gaussian primitive, the dual-branch illumination MLP regresses its final radiance $L_i(\omega_i, \mathbf{p}, t)$ and the incident light direction $\omega_i$. These outputs, together with the Disney BRDF parameters per point - specular albedo $\rho_s$, diffuse albedo $\rho_d$, and roughness $r$ - and the view direction $\omega_o$ are then substituted into Equations (4) and (5) to compute the outgoing color $C_i(t, \omega_o)$ of each Gaussian as seen by the camera. For simplicity, we define $\omega_i$ and $\omega_o$ as the unit vectors from the Gaussian center $\mathbf{p}$ to the light source and to the camera, respectively.

Finally, the per-Gaussian colors $C_i(t, \omega_o)$ and opacities $\alpha_i$ are combined via Equation 1 and rasterized following the standard 2DGS pipeline. The resulting image $I$ is supervised against ground truth $I^*$ with loss of photometric reconstruction, enabling end-to-end optimization of both Gaussian parameters and the illumination dual-branch MLP.

### 4.4 Optimization

We supervise the entire training process using the 2DGS loss:

$$L = L_c + L_d + L_n, \tag{10}$$

where $L_c$ is the photometric term, combining an $L_1$ color loss with a DSSIM component to measure appearance fidelity, $L_d$ is the *depth distortion* loss, which penalizes variance in the depths of the intersected 2D Gaussian primitives along each ray—concentrating them into a tight range to correct uneven depth distributions, and $L_n$ is the *normal consistency* loss, minimizing discrepancies between the rendered normal map and the gradient of the reconstructed depth to enforce geometric coherence. The calculation of the loss is as same as Huang et al. (2024).

## 5 Experiment

### 5.1 Implementations

All experiments were conducted on 2 NVIDIA GeForce RTX 4090 GPUs. Except for initialization-related parameters, all optimization hyperparameters were kept consistent with the 2DGS work. The

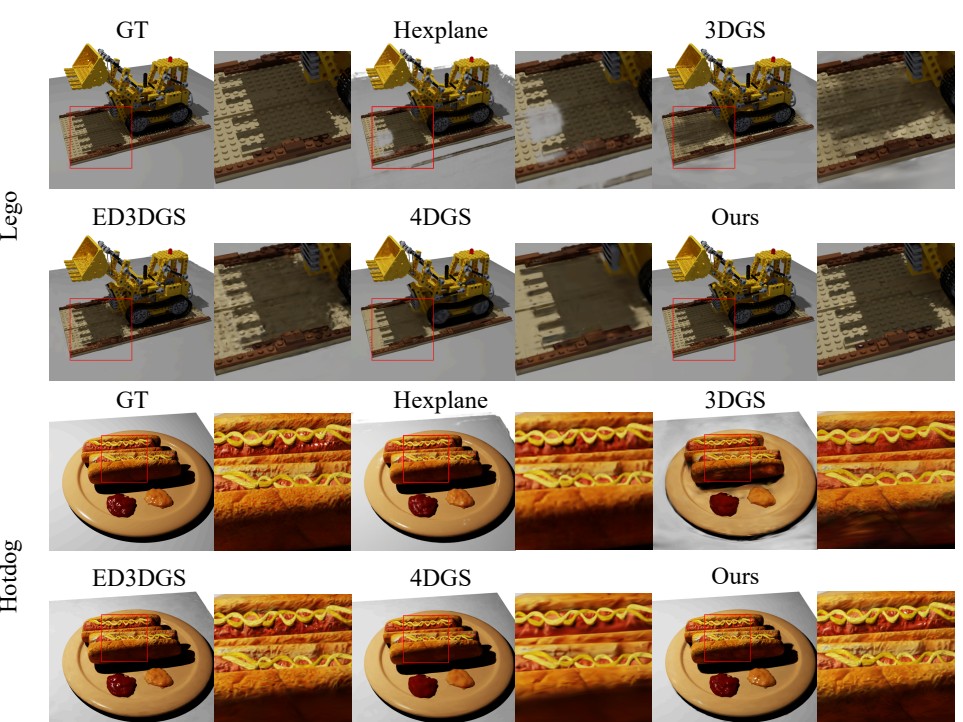

Figure 3: Qualitative comparison on two synthetic scenes.

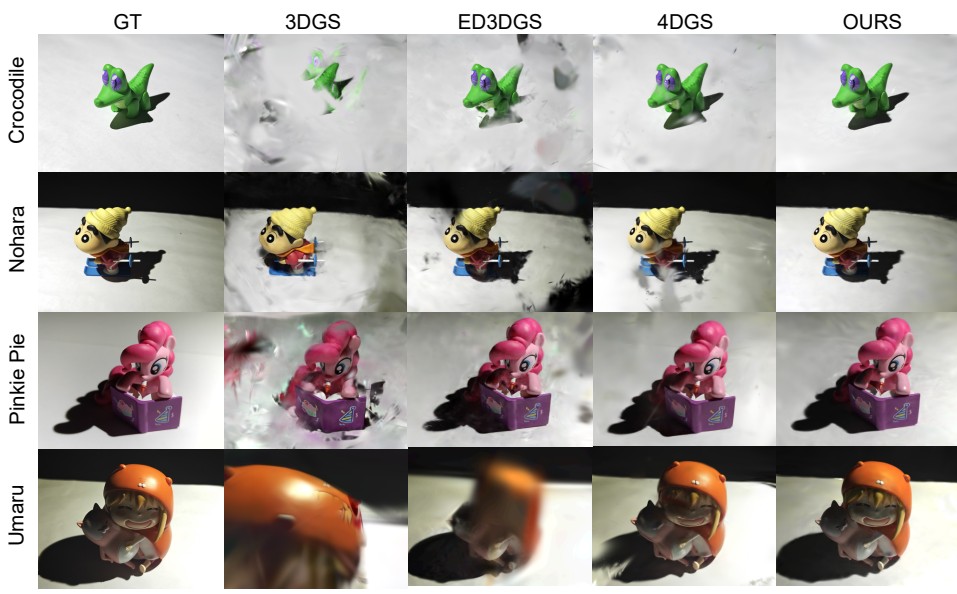

Figure 4: Qualitative comparison on real-world scenes.

learning rate for the BRDF parameters was fixed at $1.0 \times 10^{-3}$, while the dual-branch illumination MLP was trained with an initial learning rate of $1.6 \times 10^{-4}$, which is decayed to $1.6 \times 10^{-5}$ using the same step-decay schedule as in 3DGS.

**Synthetic Dataset.** Our dataset scenes consist of two components: Blender models imported from NeRF and open-source static Blender models, together with a point light source that moves continuously around the scene. The light source follows a circular trajectory at a fixed height and radius. For each scene, we sample 20 to 30 light-source positions uniformly along the circumference, and at each position we randomly render 8 images from different viewpoints, recording the corresponding camera extrinsics. Each scene thus contains approximately 160 to 240 images, and the data format matches the D-NeRF dataset Pumarola et al. (2021).

**Real-world Dataset.** For real-world evaluation, we constructed a dataset in which a single hand-held flashlight serves as the sole light source, following a casual trajectory around each scene. Notably, our real-world scenes *ARE NOT* in ideally dark conditions with only the flashlight. There is residual ambient illumination present in the environment, and ambient light reflected from the flashlight contributes additional indirect lighting. All images were captured with a smartphone whose exposure, shutter speed, and ISO sensitivity were fixed throughout the acquisition. Each scene comprises approximately 130 to 240 photographs, with 8 to 12 images taken at each discrete light-source position. Finally, we processed the captured images using COLMAP (Schönberger & Frahm, 2016; Schönberger et al., 2016) to recover the camera intrinsics and extrinsics.

## 5.2 RESULTS

We employ 3 metrics to evaluate our experimental results: peak signal-to-noise ratio (PSNR), perceptual quality (LPIPS), and structural similarity index (SSIM), which are widely used in scene reconstruction task Kerbl et al. (2023); Huang et al. (2024); Wu et al. (2024). Several representative methods of both static and dynamical scene reconstruction are compared: 3DGS Kerbl et al. (2023) and 2DGS Huang et al. (2024) are for static scene, while other methods such as 4DGS Wu et al. (2024), HexPlane Cao & Johnson (2023), ADC-GS Huang et al. (2025), DG-mesh Liu et al. (2024a), ED-3DGS Katsumata et al. (2024a) and 4DGaussians Yang et al. (2023) are

| Methods | PSNR↑ | SSIM↑ | LPIPS↓ |
|---------|-------|-------|--------|
| GS-IR | 19.52 | 0.71 | 0.38 |
| HexPlane | 25.30 | 0.74 | 0.22 |
| 3DGS | 19.44 | 0.81 | 0.25 |
| 2DGS | 20.33 | 0.85 | 0.19 |
| DG-mesh | 25.68 | 0.88 | 0.20 |
| ED-3DGS | 26.13 | 0.90 | 0.11 |
| 4DGS | 29.39 | 0.92 | 0.12 |
| **Ours** | **30.27** | **0.94** | **0.08** |

Table 1: Quantitative comparison on the synthetic dataset.

for dynamic scene, and GS-IR Liang et al. (2024), a neural inverse rendering approach.[1] Due to the inconsistent settings of dataset format, we exclud HexPlane and DG-mesh in the real-world dataset, and omit 4DGaussians and ADC-GS in the synthetic data.

The results in the synthetic data set are reported in Table 1, and those in the real world data set are reported in Table 2. All values are reproduced from the original implementations under identical parameter settings.

**Analysis.** For both synthetic and real-world datasets, static methods such as 3DGS and 2DGS struggle to reconstruct color changes caused by dynamic illumination when only using spherical-harmonic coefficients, and therefore perform poorly on both benchmarks. This effect is especially pronounced on the real-world dataset, where 3DGS overfits the training views and collapses on the held-out Umaru test scene (see Figure 4). GS-IR likewise under-performs because it assumes time-invariant illumination and thus cannot account for per-timestamp lighting changes.

| Methods | PSNR↑ | SSIM↑ | LPIPS↓ |
|---------|-------|-------|--------|
| GS-IR | 11.72 | 0.73 | 0.43 |
| 3DGS | 11.16 | 0.77 | 0.36 |
| 2DGS | 12.36 | 0.81 | 0.31 |
| ED-3DGS | 14.50 | 0.82 | 0.29 |
| 4DGaussians | 13.24 | 0.80 | 0.30 |
| ADC-GS | 22.44 | 0.92 | 0.20 |
| 4DGS | 23.25 | 0.92 | 0.22 |
| **Ours** | **25.29** | **0.94** | **0.17** |

Table 2: Quantitative comparison on the real-world dataset.

---

[1]Several recent studies Wang et al. (2024); Yuan et al. (2025) don't provide the source code. In addition, due to the characteristic of our task the source codes of several recent methods such as Bae et al. (2024); Li et al. (2024b) can not converge under the default setting in our datasets.

| Model | PSNR (dB) ↑ | SSIM ↑ | LPIPS ↓ |
|---|---|---|---|
| Ours w/o Positional Encoding | 21.09 | 0.83 | 0.20 |
| Ours w/o $\beta(\mathbf{p}, t)$ | 31.50 | 0.95 | 0.08 |
| Ours w/o $\alpha(\mathbf{p}, t)$ | 27.02 | 0.92 | 0.11 |
| **Ours** | **32.71** | **0.96** | **0.06** |

Table 3: Ablation study on the synthetic dataset.

On the synthetic dataset, dynamic methods outperform static ones because they can treat color variations, such as shadows, as object motion. By contrast, HexPlane and DG-mesh exhibit lower SSIM and higher LPIPS. The reason is that: HexPlane's implicit representation makes its errors hard to correct, while DG-mesh turns erroneous Gaussians into an overabundance of mesh elements, degrading rendering quality.

The performance gap for the same methods between synthetic and real-world datasets stems from their vastly different ambient illumination. Synthetic scenes feature almost no ambient light—most reflected rays escape rather than return to surfaces—so even as the point light moves, unobstructed objects show only imperceptible color shifts that methods tend to ignore. In contrast, real-world scenes exhibit strong ambient reflections that produce noticeable color variations on surfaces, forcing these methods to handle not only shadows but also these color variations. Methods then mistake these illumination effects for motion and spawn spurious Gaussians, degrading geometry and color learning. Techniques enforcing motion continuity (e.g., ED3DGS and 4DGaussians) struggle even more to disentangle lighting changes from object motion, resulting in poorer reconstructions.

By predicting dynamic illumination with a dual-branch MLP, assigning BRDF parameters to 2D Gaussians, and computing per-point color via the rendering equation, our method is able to accurately learn color variations under time-varying single-point lighting and faithfully reconstruct dynamically lit scenes. Experimental results show that ours approach performs well on both synthetic and real-world datasets. Additionally, the results on the real-world dataset demonstrate robustness to the residual ambient illumination present in real-world scenes and enable accurate scene reconstruction under non-ideal lighting conditions.

### 5.3 ABLATION STUDY

**Positional Encoding.** Applying positional encoding to both the 2D Gaussian coordinates and the timestamp allows the MLP to learn subtle effects—resulting in a more accurate BRDF for each Gaussian. Without positional encoding, we observe spurious Gaussians in the outputs and a significant drop in overall accuracy.

**Decoder.** From the results, the model without $g_\phi$ fails to correctly darken regions occluded from the light source and cannot capture the subtle ambient glow within shadowed areas. In contrast, when both $\beta(\mathbf{p}, t)$ and $\alpha(\mathbf{p}, t)$ are jointly learned, the network adaptively attenuates direct radiance in occluded regions and adds a smooth baseline illumination, yielding crisper shadow boundaries and preserving low-frequency ambient lighting.

## 6 CONCLUSION

We presented a novel method for dynamic scene reconstruction under unknown, time-varying single-point illumination. This approach addresses the challenging problem of dynamic single-light-source scene reconstruction by leveraging a 2D Gaussian splatting framework augmented with learned BRDF parameters. Experiments demonstrate that our method achieve superior reconstruction fidelity and rendering quality under challenging single-light conditions.

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

## A APPENDIX

## A USE OF LARGE LANGUAGE MODELS (LLMS)

**Summary.** LLMs were used as assistive tools during writing.

**Models and access.** We used ChatGPT (GPT-5 Thinking, July–Sep 2025) via the web interface. No institutional or private APIs were shared.

**Roles in the research process.**

- *Writing support.* Performed language polishing and translation of author-written content (e.g., abstract, introduction, captions). Edits were limited to grammar, clarity, and tone; the LLM did not originate technical ideas, equations, or claims.

- *Formatting and compliance.* Checked and adjusted LaTeX formatting for template compliance (ICLR class settings, tables, references and cross-references, small caps in headings, page numbering) and overall style consistency (terminology, capitalization, hyphenation). No changes to scientific content were made.

**Safeguards.**

- No proprietary data, unreleased images, or identifying reviewer information were pasted into LLM chats; prompts contained only publicly shareable text or synthetic examples.

- All mathematical statements, citations, and experimental claims from LLM drafts were checked against primary sources or re-derived by the authors; any hallucinated references were discarded.

**Accountability.** The authors are solely responsible for the accuracy and integrity of the submission. LLMs are not eligible for authorship. This disclosure follows the ICLR 2026 policy to report significant LLM usage in an appendix.

## B ADDITIONAL EXPERIMENTS

To evaluate whether temporally-varying spherical harmonic coefficients can faithfully reconstruct dynamic illumination, we introduce a time-conditioned MLP that takes as input the positionally encoded timestamp $t$ and the 3D position $(x, y, z)$ of each Gaussian, and outputs a residual update $\Delta C$ to the canonical spherical harmonic coefficients. We then compute the time-specific coefficients as

$$C'(t) = C_{\text{base}} + \Delta C, \tag{11}$$

and render colors. We implement this experiment on both the 3DGS and the 2DGS pipeline, referring to the variants as Temporal SH-3D and Temporal SH-2D, respectively. Each is trained with the same learning-rate schedule as our illumination dual-branch MLP. Qualitative results are shown in Figure 5, and quantitative comparisons appear in Table 4.

**Analysis.** As Table 4 demonstrates, both Temporal SH-3D and Temporal SH-2D underperform our proposed method on the real-world dataset. This is because, unlike in static datasets where spherical harmonic coefficients can be trained under dense view coverage, in our dynamic dataset each timestamp's coefficients are learned from only sparse viewpoints and thus cannot fully capture the color distribution at that moment. Consequently, when synthesizing novel views, these under-trained coefficients may fail to produce accurate colors, leading to degraded reconstruction quality. Moreover, acquiring sufficiently dense multi-view observations at every timestamp in a dynamic lighting scene is inherently challenging, rendering this approach impractical for reconstructing dynamic illumination. In contrast, our approach jointly supervises light-source position and intensity across all Gaussians at each timestamp and learns per-Gaussian Disney BRDF parameters under multiview supervision across all timestamps. Consequently, our method is substantially more robust and achieves higher reconstruction fidelity than both Temporal SH variants.

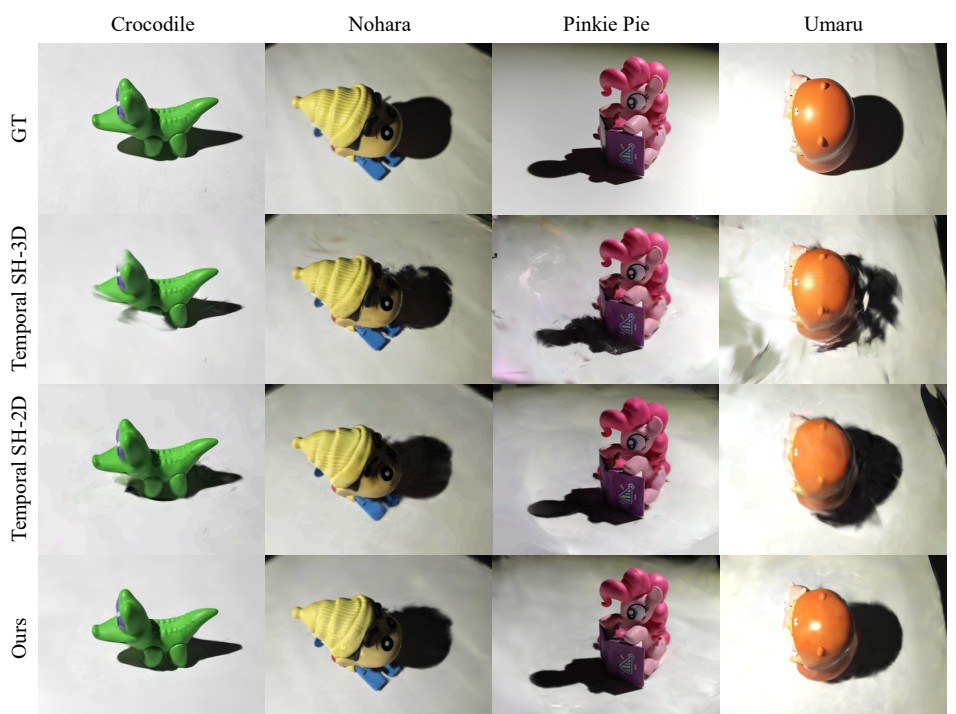

Figure 5: Qualitative comparison between the Temporal SH-3D,Temporal SH-2D and our method on real-world datasets.

| Method | PSNR | SSIM | LPIPS |
|---|---|---|---|
| Temporal SH-3D | 21.54 | 0.91 | 0.21 |
| Temporal SH-2D | 21.27 | 0.90 | 0.21 |
| Ours | 25.29 | 0.94 | 0.17 |

Table 4: Average quantitative results on all real-world scenes.

## HYPERPARAMETER SETTINGS

The illumination dual-branch MLP comprise 10 fully-connected layers. The occlusion branch's decoder is implemented as a 2-layer MLP. We employ ReLU activations throughout. Disney BRDF parameters comprise a scalar roughness $r$, a three-dimensional specular albedo $\rho_s$, and a three-dimensional diffuse albedo $\rho_d$. All inputs are encoded using a 20-dimensional positional encoding.

