# OpenReview forum: "Latent Light Source Modeling for Scene Reconstruction under Dynamic Illumination"
_ICLR.cc/2026/Conference — ICLR 2026 Conference Withdrawn Submission_

### Official Review · Reviewer_h8WZ · 2025-10-31

**Soundness:** 2
**Presentation:** 3
**Contribution:** 2
**Rating:** 2
**Confidence:** 4

**Summary:**

The paper proposes a 2D Gaussian Splatting–based framework for reconstructing scenes under unknown, time-varying single-point illumination. A dual-branch MLP estimates latent light-source position, intensity, visibility, and ambient terms, while each Gaussian carries Disney BRDF parameters to model reflectance. This formulation allows the network to disentangle illumination from material and geometry, improving reconstruction quality when lighting changes over time. Experiments on synthetic and real datasets demonstrate higher PSNR, SSIM, and lower LPIPS than recent dynamic-scene baselines such as 4DGS and ED-3DGS. The method effectively preserves shading and shadow consistency across viewpoints without requiring calibrated light information. Overall, the paper contributes a physically-motivated extension of Gaussian Splatting toward dynamic-lighting scenarios, though its practical motivation and the validation of light and material estimation remain limited.

**Strengths:**

The paper addresses an under-explored but technically challenging problem—scene reconstruction under unknown, dynamic illumination. The proposed combination of a dual-branch illumination network and BRDF-parameterized Gaussians is conceptually clean and integrates physical reasoning into the efficient 2D Gaussian Splatting framework. The formulation eliminates the need for explicit light calibration while maintaining differentiability and real-time rendering capability. Quantitative results on both synthetic and real scenes show clear gains over state-of-the-art dynamic-scene methods (4DGS, ED-3DGS) in standard reconstruction metrics. The visual comparisons reveal sharper shadow boundaries and better color stability, suggesting that the method indeed captures lighting-driven variations rather than misinterpreting them as motion. The ablation studies on positional encoding and illumination branches further clarify the contribution of each design. Overall, the technical novelty is moderate but well-motivated within Gaussian Splatting research and demonstrates solid empirical improvements.

**Weaknesses:**

The main limitations lie in motivation clarity and evaluation depth. The assumed setting—moving single-point light with otherwise dark surroundings—is artificial and rarely encountered in real applications; the paper does not convincingly articulate scenarios where such conditions matter (e.g., handheld-light scanning, robot-mounted lamps). More critically, the evaluation focuses solely on image-level reconstruction metrics. Since the method explicitly estimates light-source trajectories and per-point BRDFs, their accuracy should be validated, at least on the synthetic dataset where ground-truth light and material parameters can be recorded easily. Without such verification or any relighting demonstrations, it remains unclear whether the network truly disentangles illumination and reflectance or simply overfits appearance. Comparisons with self-calibrated photometric-stereo methods (e.g., Neural Multi-View Self-Calibrated Photometric Stereo, ICCV 2025) are also missing, even though they address nearly identical OLAT-type setups. These omissions weaken the scientific credibility of the claims.

**Questions:**

In addition to the weakness, it would be great if authors can response to the following minor comments.
- Clarify the intended application domain to justify the dynamic single-light assumption.
- Include quantitative evaluation of light-position/intensity and BRDF-parameter errors on synthetic data; such tables would substantiate the physical correctness of the model.
- Adding a small relighting experiment (rendering under novel light positions) would greatly strengthen the claim of illumination–reflectance disentanglement.
- Related-work section should mention recent uncalibrated multi-view photometric-stereo approaches (ICCV 2025 Cao et al.) as closely related.
- Some figures (e.g., Fig. 3–4) could include per-scene PSNR/SSIM labels for readability.

---

### Official Review · Reviewer_au66 · 2025-10-31

**Soundness:** 2
**Presentation:** 2
**Contribution:** 2
**Rating:** 4
**Confidence:** 4

**Summary:**

This paper presents a novel framework to model the scene under unknown dynamic lighting conditions. The proposed method parameterizes the spatiotemporally varying light source by an MLP and assigns each Gaussian with a learnable BRDF to model the material-lighting interaction. Experiment results are provided to demonstrate the effectiveness of the framework.

**Strengths:**

1. The proposed method achieves the object reconstruction under unknown dynamic lighting conditions by leveraging MLPs to model the dynamic light source and visibility of Gaussians.
2. Experiments show that the method achieves the best performance over baselines.

**Weaknesses:**

1. Perhaps an inappropriate baseline selection. Since the datasets contain the scenes under dynamic illumination, choosing baselines for static scene reconstruction, like 3DGS and 2DGS, and baselines for inverse rendering under static lighting conditions, like GS-IR, may involve unfair comparisons. From my point of view, the setting of the datasets is more like OLAT, but leaving the positions and intensities of the light sources unknown. We can easily extend the existing OLAT-based method to this setting by simply making the light source an optimizable parameter as well. Thus, the variants of GS^3 and OLAT-GS with a learnable point light source may be better baselines.

2. More ablation studies should be conducted. The proposed method leverages an MLP conditioned on the time stamp $t$ to model the light source. What if we directly optimize the position and intensity of the light source for each time stamp or use a basis function to parameterize it? As for visibility, it is conditioned on the time stamp $t$ and the Gaussian’s position $\mathbf{p}$. However, since we can obtain the position of the light source, we can directly use shadow mapping to obtain the shadow. When the light source position and geometric information are relatively accurate, shadow mapping may produce more accurate shadows. Perhaps the author could compare the performance impact of the above schemes.

3. Furthermore, the dataset settings may not be reasonable. As mentioned in the paper, for each light source, photos are captured from 8 to 12 viewpoints. Such a setup can only be achieved using a renderer like Blender or in a laboratory setting. However, in both of these cases, the scene to be reconstructed is already available, thus significantly diminishing the value of the reconstruction.

**Questions:**

1. Could you provide the comparison results on OLAT datasets, such as the NRhints dataset or the GS^3 dataset, by pretending to forget the light source information provided in the dataset?

---

### Official Review · Reviewer_84y1 · 2025-10-31

**Soundness:** 2
**Presentation:** 2
**Contribution:** 1
**Rating:** 2
**Confidence:** 5

**Summary:**

The paper presents a method to reconstruct objects where a point light source changes over time. To be more specific, the approach builds on top of 2D Gaussian splatting, augmenting each Gaussian splat with BRDF parameters, replacing the original spherical harmonics. In addition, a dual-branch MLP models the lighting of a scene, both over time and space, and predicts the parameters for the rendering.

**Strengths:**

- it makes sense to explore alternative representations for the reflectance parameters for the Gaussians. In some sense the method is somewhat a hybrid between gaussians and earlier NeRF methods.

- I appreciate the motivation of the setting where we have limited capture setups; i.e., in this case a monocular capture setting.

**Weaknesses:**

- unfortunately, the results are not convincing both in terms of quality and complexity. In particular, the setting individual, isolated objects with a single light source is highly contrived. In addition, the visuals are not that exciting - this is the main reason why I'm lukewarm on the paper.

- the method itself is technically relatively incremental over existing works. Essentially, some ideas of NeRF-based methods are integrated on top of 2DGS. While this is interesting, I would've hoped for a deeper technical exploration.

**Questions:**

Technically, I don't have that many questions - the approach is relatively clear. My major complaint is about limited results in toy scenes. In its current form, the paper should probably be re-written to object-scale with single light sources rather than dynamic scenes.

Suggested experiments:
- It might be interesting to see if the the method can be used for relighting. Here, a video with user-control might be interesting to see.
- Try out room-scale scenes with single lights. I feel that would be be minimum to be shown.

---

### Official Review · Reviewer_446t · 2025-11-01

**Soundness:** 1
**Presentation:** 3
**Contribution:** 1
**Rating:** 2
**Confidence:** 5

**Summary:**

This paper tackles the problem of 3D reconstructing a static scene under dynamically varying illumination. Once the scene gets reconstructed with appearance, one can synthesize novel views of the same scene under arbitrary user-defined illumination.

Crucially, the users opted for a latent modeling approach of the lighting, which in my understanding is representing lighting implicitly as a function of time instead of, say, the incident direction. In this setup, lighting directions are predicted by an MLP given XYZ and time t.

The foundation of this work is the 2DGS pipeline, to which the authors added material and geometry properties like albedo, roughness, and surface normals to each Gaussian. Then two MLP branches predict AO + visibility and incident direction + direct radiance, respectively, as a function also of time. The final loss is the sum of standard L1 re-rendering loss, depth distortion loss, and the normal consistency loss.

**Strengths:**

The formulation is sensible, separating time/lighting-induced variations from intrinsic material properties that remain constant across time.

The presentation is clear, making the paper easy to follow.

**Weaknesses:**

One key difference that sets apart this work from Nerfactor and other work modeling illumination is that this paper represents lighting as an implicit function of time rather than physical properties like lighting direction. This latent modeling approach blurs the line between general lighting and a single point light (i.e., OLAT style), since now the input is just time instead of the direction. However, there is no study/analysis on this key idea. Specifically, the model should be able to handle non-point lighting to some degree. When we generalize to omnidirectional lighting with a predominant direction, this model should still work, right? Showing something like this would prove the point of latent modeling.

Scene-wise, this paper is weak in that it only showed 4 OLAT real scenes, which are simplified with clean backgrounds and a centerpiece object, plus some nerf scenes. Related to my previous point, this is unsatisfying given the latent modeling claim.

Result-wise, this is also insufficient. The model predicts AO and visibility, but there’s no visualization of the learned AO and visibility. Do they make sense at all if they are visualized? Similarly, modeling lighting is largely for the relighting capability, but no result on relighting was shown? I strongly think a paper like this should show relighting *videos*.

The whole visibility caching idea was a key contribution of NeRV, which the authors didn’t even cite.

**Questions:**

Is it true that the model works to some degree dealing with non-point lighting due to its latent modeling approach?

Do AO and visibility learned make sense when visualized?

Have the authors tried this on more realistic scenes?

Have the authors tried relighting?

---

### Note · Authors · 2025-11-12

I have read and agree with the venue's withdrawal policy on behalf of myself and my co-authors.